# Cholesterol Metabolism in Antigen-Presenting Cells and HIV-1 Trans-Infection of CD4^+^ T Cells

**DOI:** 10.3390/v15122347

**Published:** 2023-11-29

**Authors:** Daniel Okpaise, Nicolas Sluis-Cremer, Giovanna Rappocciolo, Charles R. Rinaldo

**Affiliations:** Department of Medicine, Division of Infectious Diseases, University of Pittsburgh School of Medicine, Pittsburgh, PA 15261, USA; dmo52@pitt.edu (D.O.); nps2@pitt.edu (N.S.-C.); giovanna@pitt.edu (G.R.)

**Keywords:** HIV-1, antiretroviral therapy, trans-infection, cholesterol metabolism, antigen-presenting cells, immunometabolism, latent reservoir, non-progressor

## Abstract

Antiretroviral therapy (ART) provides an effective method for managing HIV-1 infection and preventing the onset of AIDS; however, it is ineffective against the reservoir of latent HIV-1 that persists predominantly in resting CD4^+^ T cells. Understanding the mechanisms that facilitate the persistence of the latent reservoir is key to developing an effective cure for HIV-1. Of particular importance in the establishment and maintenance of the latent viral reservoir is the intercellular transfer of HIV-1 from professional antigen-presenting cells (APCs—monocytes/macrophages, myeloid dendritic cells, and B lymphocytes) to CD4^+^ T cells, termed trans-infection. Whereas virus-to-cell HIV-1 cis infection is sensitive to ART, trans-infection is impervious to antiviral therapy. APCs from HIV-1-positive non-progressors (NPs) who control their HIV-1 infection in the absence of ART do not trans-infect CD4^+^ T cells. In this review, we focus on this unique property of NPs that we propose is driven by a genetically inherited, altered cholesterol metabolism in their APCs. We focus on cellular cholesterol homeostasis and the role of cholesterol metabolism in HIV-1 trans-infection, and notably, the link between cholesterol efflux and HIV-1 trans-infection in NPs.

## 1. The Importance of HIV-1 Trans-Infection of CD4^+^ T Cells in Maintaining the HIV-1 Reservoir during Antiretroviral Therapy (ART)

ART provides an effective method for managing HIV-1 and can be used to protect those at high risk of infection. While an effective ART regimen reduces the plasma HIV-1 load to low or undetectable levels and stops progression to AIDS [1,2], it does not cure HIV-1 infection. A persistent reservoir of latent HIV-1 that resides predominantly in resting CD4^+^ T cells represents a major obstacle to eradicating replication-competent HIV-1 in people with HIV-1 (PWH). HIV-1 can infect CD4^+^ T cells via two major pathways: (1) cis-infection, a pathway in which cell-free viruses directly infect CD4^+^ T cells, and (2) trans-infection, where cell-to-cell HIV-1 transmission is mediated by CD4^+^ T cells or professional antigen-presenting cells (APCs), i.e., myeloid dendritic cells (DCs), monocytes/macrophages, and B lymphocytes (Figure 1). Remarkably, HIV-1 trans-infection mediated by APCs is 10 to 1000-fold more efficient at infecting CD4^+^ T cells compared to cis-infection in vitro [3,4,5,6,7,8], and trans-infection can shield HIV-1 from immune surveillance, neutralizing antibodies, and ART [9,10,11,12,13,14]. The three major types of APCs vary in their efficiency at mediating HIV-1 trans-infection [15,16,17]. However, despite major differences in myeloid and lymphoid lineage and immune function, all three types of APCs lead to HIV-1 trans-infection via a type II C-type lectin receptor DC-specific ICAM-3 grabbing non-integrin (DC-SIGN; CD209) [15], whereas monocytes/macrophages and myeloid DCs but not B lymphocytes also express the lectin Siglec-1 (CD169) that drives HIV-1 trans-infection [16,17,18,19]. These lectins facilitate APC capture of the virus, internalization in intracellular compartments called multivesicular bodies [20,21], and transfer to a CD4^+^ T cell via the virological synapse. Furthermore, the interaction between HIV-1 and Siglec-1 in DCs/macrophages has been shown to induce the formation of virus-containing compartments, which allow for the trafficking of the virus through the synapse into the CD4^+^ T cells [21]. The internalization of the virus in these virus-containing compartments or multivesicular bodies shields the virus from neutralizing antibodies or other immune defenses such as HIV-1-specific CD8^+^ T cells. Typically, APCs loaded with HIV-1 can trans-infect different populations of CD4^+^ T cells, including activated and memory subsets [22,23].

## 2. Insights Provided by HIV-1 Non-Progressors (NPs) on the Importance of APC-Mediated HIV-1 Trans-Infection of CD4^+^ T Cell Subsets and the Impact on the Establishment and Maintenance of the HIV-1 Reservoir

Approximately 5% of PWH are NPs who control HIV-1 disease progression in the absence of ART. NPs are a heterogeneous group typically characterized by consistently low to undetectable HIV-1 loads and a CD4^+^ T cell count of >500/mm^3^ [24,25,26]. Mechanisms associated with the NP phenotype include defective HIV-1 variants, increased cytotoxic CD8^+^ T cell killing of HIV-1-infected CD4^+^ T cells, CCR5 receptor mutations, human leukocyte antigen types, neutralizing antibodies, insertion of provirus into genetic deserts, and apolipoprotein B mRNA editing enzyme catalytic subunit 3G production [24,25,26].

We reported that APCs from NPs lack the capacity to HIV-1 trans-infect CD4^+^ T cells in vitro and that this phenotype is driven by altered cholesterol metabolism and decreased cellular and lipid rafting in APC membranes [26]. Interestingly, our group recently demonstrated that B lymphocytes are highly efficient at trans-infecting naïve CD4^+^ T (T_N_) cells [16]. In line with this finding, CD4^+^ T_N_ cells from NPs do not harbor latent replication-competent HIV-1, likely due to inefficient APC-mediated trans-infection linked to their altered cell cholesterol metabolism [27,28,29]. This is critical to understanding the latent viral reservoir, as CD4^+^ T_N_ cells, unlike activated or other resting CD4^+^ T cells, do not express CCR5. As such, CD4^+^ T_N_ cells are resistant to infection by R5-tropic HIV-1 [30,31]. Indeed, Pinzone et al. reported that CD4^+^ T_N_ cells are a key contributor to maintaining the intactness of the latent viral reservoir before and after the initiation of ART [28]. Following ART initiation, CD4^+^ T_N_ cells contribute 8% to 59% of the latent HIV-1 reservoir, and after four to eight years of ART, this value ranges from 20% to 34% [29]. This highlights the important role of CD4^+^ T_N_ cells in viral persistence, reservoir size, and diversity [22,29].

## 3. Cellular Cholesterol Homeostasis: An Intricate, Balanced, and Tightly Controlled Process That, If Perturbed, Can Dysregulate Key Cellular Functions

Cholesterol uptake is mediated by the transport of extracellular cholesterol from the plasma membrane into cells by low-density lipoprotein receptor-mediated endocytosis of low-density lipoprotein particles (Figure 2 Steps A–C) [32,33,34,35]. Endocytosed cholesterol is then intracellularly distributed by Niemann–Pick type C 1 and 2 proteins to the endoplasmic reticulum and plasma membrane (Figure 2 Steps D–E) [36,37,38,39]. Cells can synthesize their own cholesterol via a regulated cascade involving cytoplasmic enzymes in the endoplasmic reticulum and peroxisomes. They also store excess cholesterol, which is toxic, via esterification by acyl-coenzyme A: cholesterol acyltransferase (Figure 2, Step E) [40]. Upon esterification, esterified cholesterol is stored as lipid droplets or exported via reverse cholesterol transport mediated by efflux genes such as ATP-binding cassette A1 (ABCA1) or ATP-binding cassette G1 (ABCG1) [41].

Cholesterol homeostasis is central to cell function. This homeostasis is controlled by two transcription factors: sterol response element-binding protein (SREBP) and nuclear receptors, liver X receptors (LXR), and peroxisome proliferator-activated receptor gamma (PPARγ). Each transcription factor controls an opposing transcriptional role (Figure 2) [41]. During cholesterol depletion in the cell, the sterol response element-binding protein is activated, which is escorted by a sterol response element-binding protein 2 cleavage-activating protein from the endoplasmic reticulum to the Golgi. There, cleavage occurs, mediated by site 1 protease and site 2 protease. Cleavage leads to the formation of the mature form of SREBP2, which binds to the sterol regulatory element in the adjacent region of the promoter for 3-hydroxy-3-methylglutaryl-CoA reductase (Figure 2 Steps B*–C**). This binding mediates the transcription of 3-hydroxy-3-methylglutaryl-CoA reductase, thus speeding up cholesterol synthesis to restore normalcy in cellular cholesterol levels [42,43].

Activation of SREBP2 leads to the transcription of the LDL receptor gene, which is processed through the endoplasmic reticulum, leading to an increased expression of LDL receptors on the plasma membrane (Figure 2 Step C*). In contrast, during elevated cellular cholesterol levels, SREBP is repressed in the endoplasmic reticulum, and liver X receptors (LXR) are activated by oxysterol, which leads to a downstream induction of genes repressing cholesterol uptake and efflux-mediating genes such as *ABCA1*/*ABCG1* (Figure 2 Steps I–III) [41,42].

LXRs are described as master regulators for cholesterol due to their ability to sense excess cholesterol and upregulate sterol transporters such as ATP-binding cassette (ABC) genes such as *ABCA1* and *ABCG1.* Expression of *ABCA1* and *ABCG1* drives the efflux of excess cholesterol molecules out of the cell to high-density lipoprotein particles with Apo acceptors such as ApoA1 (Figure 2 Steps IV–VI) [43]. Activation of LXRs further reduces cholesterol and LDL uptake into the cell by inducing the expression of a degrader of the LDL receptor [44].

Of the cholesterol pathways detailed above, ABCA1 and ABCG1, which are involved in both cholesterol uptake and homeostasis, are critically important for APC-mediated HIV-1 trans-infection. As described below, upregulation of *ABCA1* and *ABCG1* in APCs induces free cholesterol efflux that reduces the cell’s lipid rafts and downregulates the intra-membrane receptors required for HIV-1 trans-infection. NPs’ APCs, but not CD4^+^ T cells, express higher levels of *ABCA1*, and this difference in expression is not random or HIV-1-driven but rather an intrinsic genetic trait unique to all NPs.

Cholesterol is an important energy source for many immune cell processes, including dividing and acquiring effector functions [45,46]. Cholesterol metabolism plays a central role in cells transitioning from a quiescent state to an active state through an upregulation in cholesterol and fatty acid biosynthesis [47,48]. Cellular cholesterol and the levels of cholesterol in the APC plasma membrane are critical for the expression of certain cellular cytokine markers and the localization of pattern recognition receptors [36,37]. Therefore, understanding how the alterations in APC cholesterol levels affect the expression of these cellular receptors and lipid rafts is key to determining how cholesterol metabolism affects HIV-1 trans-infection.

## 4. Cholesterol Efflux and HIV-1 Trans-Infection in NPs: Cholesterol Efflux Is One-Half of the Intricate Process of Cholesterol Metabolism for NP APCs Mediating HIV-1 Trans-Infection

We propose that cholesterol plays a critical role in establishing virological and immunological synapses between APCs and CD4^+^ T cells. The role of cholesterol metabolism in trans-infection, therefore, requires deeper exploration to unravel the mechanism by which it modulates the transfer of HIV-1 from APCs to CD4^+^ T cells. Lipid rafts, which are cholesterol-rich domains, influence the distribution and dynamics of key membrane proteins such as MHC II and ICAM-1, which are important for establishing synaptic interactions with CD4^+^ T cells [47,48,49] (Figure 3). An interference or reduction in lipid raft levels would lead to a reduction in avidity between APC and CD4^+^ T cells, thus disrupting key lymphocyte signaling required for the establishment of the synapse for virus transfer. We hypothesize that cholesterol plays a central role in the intracellular trafficking of the virus to the synapse via multivesicular bodies/virus-containing compartments [50,51,52,53]. APCs (B lymphocytes, myeloid dendritic cells, and monocytes/macrophages) utilize lectin receptors such as DC-SIGN and Siglec-1 for virus uptake. The virus is then internalized in an intracellular vesicle and trafficked to the CD4^+^ T cell via the virological synapse using a lipid vesicle (see Figure 3). An increase in the efflux of cholesterol and subsequent depletion of the lipid raft, as observed in NP APCs, alters the trafficking process and affects the distribution of membrane proteins required to establish the synapse.

Increased cholesterol efflux in APCs isolated from NPs inhibits HIV-1 trans-infection [7,23], while reconstitution of the cholesterol content in membranes from NP myeloid DCs and B cells restores their capacity to HIV-1 trans-infect CD4^+^ T cells [23]. We propose that reducing the total cellular cholesterol in APCs due to increased efflux of cholesterol drives a reduction in the membrane lipid rafts, thus altering the formation of virological synapses during APC–CD4^+^ T cell interactions. We further hypothesize that cholesterol metabolism in APCs of progressors is quite different due to the equilibrium between cholesterol biosynthesis and efflux. Maintenance of this equilibrium leads to conserving lipid rafts in the plasma membrane and free cholesterol particles within APCs.

β-Hydroxy β-methylglutaryl-CoA (HMG-CoA) reductase inhibitors, i.e., statins, are drugs that lower cholesterol and promote downregulation of glycolytic and oxidative metabolism, which affect the downstream reprogramming of immune cells [48]. Inhibition of cholesterol metabolism by statins reduces the ability of APCs to HIV-1 trans-infect CD4^+^ T cells from PWH [23]. In fact, HIV-1 trans-infection from DCs or B cells to autologous CD4^+^ T cells is reduced by 60% and 90%, respectively, following lovastatin treatment [23]. The role of APC cholesterol in mediating HIV-1 trans-infection is further highlighted using B cells from people without HIV-1 (PWOH). Thus, removing membrane cholesterol using lipophilic cyclic oligosaccharide methyl-β-cyclodextrin (BCD) from B cells from PWOH prior to co-culture with CD4^+^ T cells inhibits HIV-1 trans-infection [23]. Collectively, these results underscore the critical role that cholesterol plays in APC-to-CD4^+^ T cell transfer of HIV-1 and highlight that cholesterol is both central for immune signaling and effector function through metabolic reprogramming of APCs and critical for APC-mediated HIV-1 trans-infection of CD4^+^ T cells [23,26].

Thus, an inability to form virological synapses in NPs prevents cell-to-cell transfer of HIV-1 infection. APCs isolated from NPs possess unique perturbations in efflux genes that are not observed in CD4^+^ T cells [23,26]. These result in a significant decrease in the cholesterol content in APCs of NPs compared to HIV-1 progressors or PWOH, whereas there are similar levels of cholesterol in the CD4^+^ T cells isolated from NPs, progressors, and PWOH. mRNA expression data from APCs isolated from NPs reveal a higher expression of *ABCA1* than APCs from progressors [23]. Knockdown of *ABCA1* expression by siRNA in APCs isolated from NPs restores their capacity to HIV-1 trans-infect autologous CD4^+^ T cells. These findings highlight cholesterol efflux genes such as *ABCA1* as critical determinants of HIV-1 trans-infection.

Several studies have emphasized the role of cholesterol in APC-mediated HIV-1 trans-infection of CD4^+^ T cells [23,25,54,55,56,57,58]. DCs treated with nuclear receptor ligands targeted at the peroxisome proliferator-activated receptor (PPARγ) and LXR, both of which induce upregulation of *ABCA1* and *ABCG1,* result in increased cholesterol efflux and inhibited transfer of HIV-1 to CD4^+^ T cells [23]. There is also an observed increase in the expression of PPARγ in macrophages from NPs [26]. APCs from HIV-1-exposed seronegative individuals who are resistant to HIV-1 infection express cholesterol efflux genes at higher levels compared to healthy controls [59]. Importantly, these observed alterations in the expression of *ABCA1* in APCs of NPs are not random or virus-driven but intrinsic genetic traits highlighted by the unique disease state. Data from pre and post-HIV-1 seroconversion B cells of NPs reveal their inability to HIV-1 trans-infect CD4^+^ T cells at both time points [23]. This is supported by APCs from certain PWOHs having elevated cholesterol metabolism and being unable to trans-infect CD4^+^ T cells [25]. These observations underline the critical role of these cholesterol metabolism alterations in NPs being heritable.

This research underscores the importance of the unique, genetically encoded alterations to APC cholesterol metabolism in preventing HIV-1 trans-infection. By unraveling these genetic differences, such as host genetic markers, the occurrence of single nucleotide polymorphisms, and their distribution in the general populace, could be found to be critical to controlling and eliminating the HIV-1 reservoir in PWH. In addition, HIV-1-exposed seronegatives display a natural resistance to trans-infection by HIV-1 in vitro [60]. Moreover, there is a marked upregulation in the expression of genes associated with cholesterol efflux and interferon I activity compared to healthy controls, thereby supporting the concept of an intrinsic and heritable cholesterol metabolism-driven prevention of trans-infection.

## 5. Conclusions

We propose that alterations to the cholesterol metabolism pathway in APCs are critical to the capability of NPs to limit their HIV-1 reservoir. Therefore, there is a need to unravel the functions of this pathway in relation to immune signaling, effector function, and inflammation with a view to understanding how it enables NPs to limit disease progression. In addition, identifying single nucleotide polymorphisms and variations associated with this phenomenon of inhibition of trans-infection observed in NPs and its effect on cholesterol metabolism would be informative for recapitulation in PWH. Applying these findings to PWOH could designate those with the innate trait of resistance to HIV-1 trans-infection found in NPs. Understanding the mechanisms by which cholesterol metabolism in APCs of NPs mediates resistance to trans-infection could be pivotal to future therapeutics for reducing reservoir establishment in PWH.

Here, we considered various components of the cholesterol metabolism pathway and how an alteration in each could either promote HIV-1 trans-infection or lower it. Furthermore, this offers a clearer perspective of how cholesterol biosynthesis inhibitors such as statins function and might be optimized for more effective control of trans-infection and subsequent disease progression. We propose that specific alterations to the expression of levels of distinct genes within the cholesterol metabolism pathway within APCs of NPs hold the key to addressing the problem of controlling the latent reservoir during ART.

## Figures and Tables

**Figure 1 viruses-15-02347-f001:**
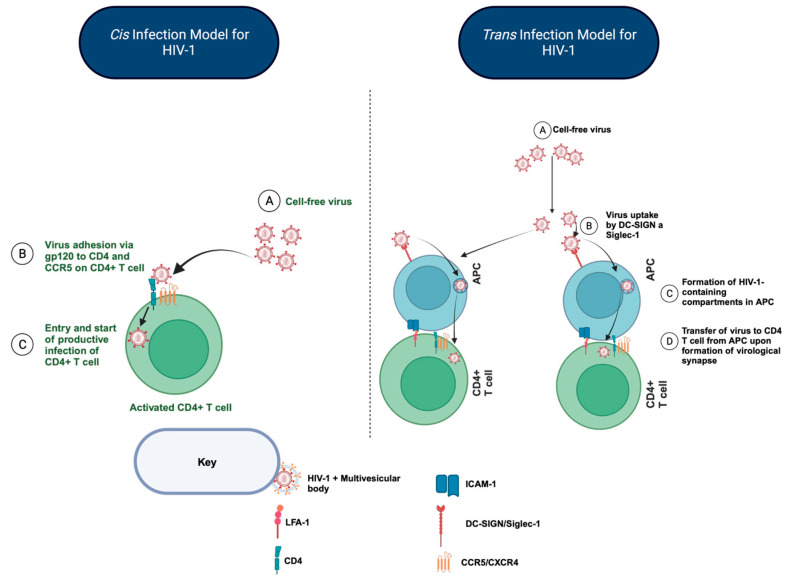
A schematic representation of the cis-infection and trans-infection models of HIV-1. (1) During cis-infection, cell-free viruses are transmitted via a site of infection, e.g., mucosal, and then infect activated CD4^+^ T cells by specific binding of the viral gp120 glycoprotein to cellular receptor CD4 in the presence of chemokine receptors such as CCR5. (2) In trans-infection, cell-free viruses are taken up by professional APCs (monocytes/macrophages, B lymphocytes, and myeloid dendritic cells) via DC-SIGN and Siglec-1. Upon virus uptake, the virus is internalized in multivesicular bodies and then transferred by the trafficking of these multivesicular bodies to the virological synapse formed during APC-CD4^+^ T cell interactions.

**Figure 2 viruses-15-02347-f002:**
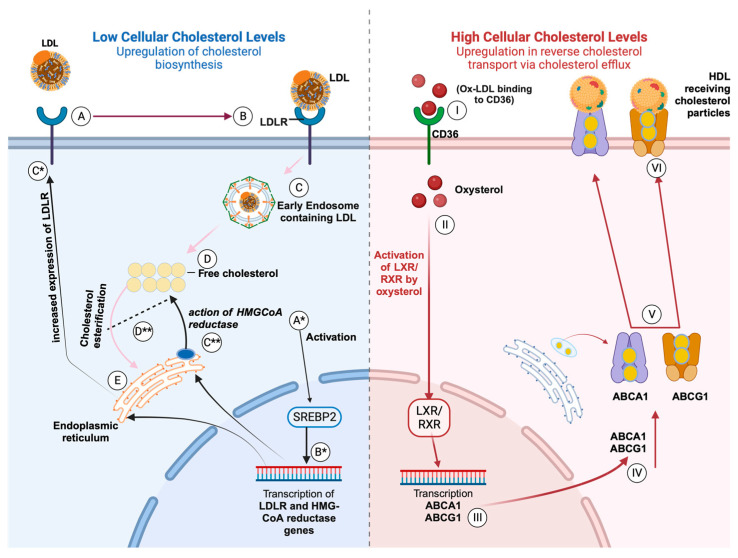
Cholesterol metabolism pathway illustrating the transcriptional control of cholesterol biosynthesis and reverse cholesterol transport by SREBP and LXR/RXR, respectively. **Steps A–E and A*–D**** describe the process of cholesterol biosynthesis during low cellular cholesterol levels and are regulated by SREBP. This involves the synthesis of new cholesterol by HMG-CoA reductase (**steps A*–C****) or influx of cholesterol/LDL particles via the LDL receptor (**steps A–E and B*–C***). **Steps I–VI** describe reverse cholesterol transport, which is regulated by LXR/RXR during high cellular cholesterol levels, signaled by the presence of oxysterol. The binding of oxysterols to LXR/PPARγ leads to their activation and subsequent upregulation in cholesterol efflux genes such as *ABCA1* and *ABCG1* and the upregulation of CD36 which further modulates cholesterol efflux.

**Figure 3 viruses-15-02347-f003:**
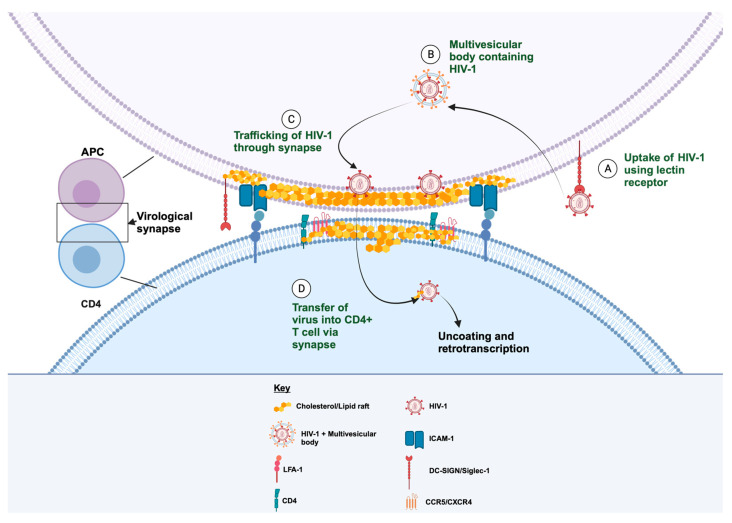
A schematic representation of trans-infection via the formation of a virological synapse through the interaction between cellular receptors and integrins. (**A**) Uptake of cell-free virus by APCs using DC-SIGN or Siglec-1. (**B**) Internalization of HIV-1 in the cholesterol-rich multivesicular body. (**C**,**D**) Trafficking of the virus to the plasma membrane for transfer via the virological synapse.

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
