# Peer review of "Cholesterol Metabolism in Antigen-Presenting Cells and HIV-1 Trans-Infection of CD4+ T Cells"

_viruses, 2023, doi:10.3390/v15122347_

Round 1

Reviewer 1 Report

Comments and Suggestions for Authors

This review article is very well written and raises some interesting hypotheses. Perhaps increase the quality of Figure 1; the font is small. I would have liked a figure summarizing the trans and cis infection hypothesis between non-progressors versus progressors.  

Author Response

- We increased the fonts and labels as shown in Figure 2.

- Furthermore, a schematic diagram was developed to show HIV-1 trans and cis infection models. We believe this addresses the reviewer’s comments and summarizes our hypothesis on HIV-1 cis and trans infection.

Reviewer 2 Report

Comments and Suggestions for Authors

This review by Okpaise and colleagues was nice and succinct. They bring to light the role of cholesterol mentalism in APCs and how that affects the possibility of trans HIV infection. While the authors focused a bit more on cholesterol metabolism for non-progressors, it is also important in the highly debated mechanisms that lead to naïve CD4+ T cell infection. The review was well written and only had a few specific points to bring up that would be helpful if the authors could comment on.

Lines 58-61: The statement on why naïve CD4+ T cells do not harbor replication-competent proviruses is flawed. We know the fraction of intact proviruses in naïve CD4+ T cells is smaller than other memory subsets. So, in the case of the lower estimate of ~1/million being replication-competent, just has a lower likelihood of detecting them in this cell population and may have nothing to do with trans infection. This is continued with the following statement about non-progressors. The level of HIV-1 DNA in NP vs CPs is ~20-30-fold fewer by PBMC measurement. It might not be possible to adequately sample with enough power.

Line 62-63: Anatomical location of naïve CD4 T cells also guides expression of CCR5. For instance, 5% of circulating CD4 express CCR5 whereas in the gut up to 60%.

Line 198-199: This might be too strong of a statement. Predominant HIV infection will still occur as virus free. I think an argument could be made to specify trans infection in certain anatomical sites (e.g., germinal centers).

Specific questions:

1)    What would be the authors hypothesis on cholesterol metabolism in chronic progressors?

2)    It is also possible that I didn’t fully make the connection, so my apologies. Maybe it is too early to say anything definitive, but it felt that the review “lacked” an actual proposed mechanism to tie together APC trans infection to CD4 and how it relates to cholesterol. All the parts are in the review, but a direct stated hypothesis could be useful.

Author Response

  1. While the mechanism by which cholesterol metabolism modulates trans infection in chronic progressors, we propose a mechanism by which this happens in Lines 153-157.

The role of cholesterol metabolism in trans infection therefore requires a deeper exploration to unravel the mechanism by which it modulates the transfer of HIV-1 from APCs to CD4+ T cells. Lipid rafts, which are cholesterol-rich domains, influence the distribution and dynamics of key membrane proteins such as MHC II and ICAM-1 which are important for establishing synaptic interactions with CD4+ T cells [47–49] (Fig. 3).”

  1. In addition, we share a mechanism that connects HIV-1 trans infection from APCs to CD4+ T cells and the role of cholesterol metabolism in Lines 152-168. Also, we state a direct hypothesis on how cholesterol metabolism modulates trans infection of HIV-1 from APC to CD4+ T cells as described in Lines 158-160.

An interference or reduction in lipid raft levels would lead to a reduction in avidity between APC and CD4+ T cells, thus disrupting key lymphocyte signaling needed for the establishment of the synapse for virus transfer. We hypothesize that cholesterol plays a central role in the intracellular trafficking of the virus to the synapse via multivesicular bodies/virus-containing compartments [50–53].”

Reviewer 3 Report

Comments and Suggestions for Authors

Comments for the review by Okpaise et al.

The authors reviewed HIV trans infection of CD4+ T cells through transfer of the virus from antigen-presenting cells, including macrophages, dendritic cells and B cells.  The manuscript is mostly well written.  I have some comments about the presentation in the paper that can be improved to make the model clear to the audience.

1. Line 157: The authors should elaborate on virological synapses.  What cell types, cell surface receptors are involved here during HIV trans infection?  Drawing a diagram will help the readers to better understand the authors’ models.  

2. Figure 1 illustrates the low and high cholesterol levels in the cells.  This seems to be a little detached from the main point in this review.  How do cholesterol levels affect trans infection?  Adding how the cholesterol metabolism affects HIV trans infection of T cells in the diagram will be helpful.

Comments on the Quality of English Language

The manuscript is mostly well written.

Author Response

  1. Figure 3 provides a schematic representation of how virological synapses are formed between APC and CD4+ T cells while showing key cellular surface receptors and cell types involved.
  2. We also offer a descriptive mechanism through which cholesterol levels modulate trans infection of HIV-1 from APCs to CD4+ T cells as described in Linea 153-162.

“The role of cholesterol metabolism in trans infection therefore requires a deeper exploration to unravel the mechanism by which it modulates the transfer of HIV-1 from APCs to CD4+ T cells. Lipid rafts, which are cholesterol-rich domains, influence the distribution and dynamics of key membrane proteins such as MHC II and ICAM-1 which are important for establishing synaptic interactions with CD4+ T cells [47–49] (Fig. 3). An interference or reduction in lipid raft levels would lead to a reduction in avidity between APC and CD4+ T cells, thus disrupting key lymphocyte signaling needed for the establishment of the synapse for virus